# Model for a novel membrane envelope in a filamentous hyperthermophilic virus

Peter Kasson[1,2], Frank DiMaio[3], Xiong Yu[4], Soizick Lucas-Staat[5], Mart Krupovic[5], Stefan Schouten[6,7], David Prangishvili[5]*, Edward H Egelman[4]*

[1]Department of Molecular Physiology and Biological Physics, University of Virginia, Charlottesville, United States; [2]Department of Biomedical Engineering, University of Virginia, Charlottesville, United States; [3]Department of Biochemistry, University of Washington, Seattle, United States; [4]Department of Biochemistry and Molecular Genetics, University of Virginia, Charlottesville, United States; [5]Department of Microbiology, Institut Pasteur, Paris, France; [6]NIOZ Royal Netherlands Institute for Sea Research, Texel, Netherlands; [7]Department of Marine Microbiology and Biogeochemistry, Utrecht University, Texel, Netherlands

**Abstract** Biological membranes create compartments, and are usually formed by lipid bilayers. However, in hyperthermophilic archaea that live optimally at temperatures above 80°C the membranes are monolayers which resemble fused bilayers. Many double-stranded DNA viruses which parasitize such hosts, including the filamentous virus AFV1 of *Acidianus hospitalis*, are enveloped with a lipid-containing membrane. Using cryo-EM, we show that the membrane in AFV1 is a ~2 nm-thick monolayer, approximately half the expected membrane thickness, formed by host membrane-derived lipids which adopt a U-shaped 'horseshoe' conformation. We hypothesize that this unusual viral envelope structure results from the extreme curvature of the viral capsid, as 'horseshoe' lipid conformations favor such curvature and host membrane lipids that permit horseshoe conformations are selectively recruited into the viral envelope. The unusual envelope found in AFV1 also has many implications for biotechnology, since this membrane can survive the most aggressive conditions involving extremes of temperature and pH.

*For correspondence: david. prangishvili@pasteur.fr (DP); egelman@virginia.edu (EHE)

## Introduction

Many viruses, including some of the most devastating human pathogens such as Ebola virus, are enveloped with a lipid membrane. The membrane is considered to be an adaptation to the host that has been convergently acquired in different virus orders (*Buchmann and Holmes, 2015*). Some groups of evolutionarily related viruses contain both enveloped and non-enveloped members. One example is provided by hyperthermophilic archaeal viruses of the order *Ligamenvirales* (*Prangishvili and Krupovic, 2012*) which contain non-enveloped, rigid rod-shaped viruses of the family *Rudiviridae* and enveloped, flexible filamentous viruses of the family *Lipothrixviridae*. Viruses from these two families have many homologous genes and build their virions using structurally similar major capsid proteins. The ~4 Å-resolution cryo-EM structure (*DiMaio et al., 2015*) of a less-complex rudivirus, the *Sulfolobus islandicus* rod-shaped virus 2 (SIRV2) (*Prangishvili et al., 1999*), has shown that the viral double-stranded (ds) DNA is completely insulated from the external medium by the single capsid protein which transforms the viral DNA into A-form, explaining the stability of rudiviruses in extremely aggressive (80°C, pH 3) natural habitats. Virions of *lipothrixviruses* are more complex and are built from two paralogous major capsid proteins (MCP1 and MCP2) which bind dsDNA to form the nucleocapsid (*Goulet et al., 2009*). The nucleocapsid is enveloped by a lipid

**eLife digest** Virtually every environment on the planet is home to some form of life, even places that, at first glance, appear to be too harsh for any organism to survive in. For example, a microscopic organism known as *Acidianus hospitalis* thrives in highly acidic environments that are hotter than 80°C, conditions that would kill humans and many other species.

*Acidianus hospitalis* has many adaptations that allow it to survive in its extreme environment. For example, the membrane that surrounds its cells has a different structure to the membranes that surround the cells of most other species. Membranes are made of molecules known as lipids. Generally these lipids assemble into two distinct layers (known as a bilayer) to form the membrane. However, in *A. hospitalis* the membrane contains only a single layer of lipids that resembles a bilayer in which lipids in opposite layers have fused together to make longer molecules.

A virus known as AFV1 is able to infect *A. hospitalis*. Like many other viruses, AFV1 steals part of its host cell's membrane when it leaves the cell in search of new cells to infect. This stolen membrane helps to protect the virus from its surroundings, however, the structure of the membrane surrounding AFV1 was not known.

Kasson et al. combined a technique called cryo-electron microscopy with computer simulations to study the membrane surrounding AFV1. The study shows that this membrane is only half as thick as the membrane that surrounds *A. hospitalis*. To make this thinner membrane, flexible lipid molecules from the *A. hospitalis* membrane bend into a U-shape.

These findings reveal a new type of membrane structure not previously seen in the natural world. In the future, this thinner membrane could have many uses in biotechnology, such as to make probes for medical imaging in patients or to deliver drugs to specific sites in the body. Enveloped by this unusual membrane, these structures may be more resistant to the normal processes that degrade and destroy foreign materials in humans and other organisms.

membrane and the termini of the virion are decorated with specialized structures involved in host recognition (*Bettstetter et al., 2003*).

Membranes of hyperthermophilic archaea often consist of unusual tetraether lipids, which form monolayers rather than bilayers typical of bacterial and eukaryotic membranes (*De Rosa and Gambacorta, 1988*; *Valentine, 2007*). It has been demonstrated that such membranes are more rigid and stable than typical bilayers, a property with considerable biotechnological appeal. However, molecular details of membrane remodeling in archaea remain grossly understudied, and how a virus captures a non-bilayer membrane from its host is virtually unknown and is conceptually difficult to reconcile with our current understanding of membrane remodeling by viruses infecting eukaryotes (*Harrison, 2008*, *2015*; *Hurley and Hanson, 2010*). Comparison of the organization of rudiviruses and lipothrixviruses provides an opportunity to probe structural changes permitted by envelope acquisition and can also explain how the lipid envelope may contribute to resisting stresses posed by the extreme geothermal environment.

## Results

To better understand the evolutionary relationship between enveloped and non-enveloped viruses, we used cryo-EM (*Figure 1*) to determine the structure of *Acidianus* filamentous virus 1 (AFV1), the prototypical lipothrixvirus infecting the hyperthermophilic and acidophilic archaeon *Acidianus hospitalis* (*Bettstetter et al., 2003*), and compared the resultant structure to that of SIRV2.

Determining the structure of AFV1 was complicated by the fact that the virions are significantly more flexible, both with respect to bending as well as extension and compression, than those of SIRV2. Segments could be classified by the pitch of the prominent helix which ranged from 39 to 47 Å (*Figure 2*), in contrast to the rather fixed pitch of 42 Å in SIRV2 (*DiMaio et al., 2015*). A three-dimensional reconstruction (*Figure 3*) of AFV1 not only reveals the gross morphology but also has allowed us to build a full atomic model for both the two MCP subunits and the DNA. While the Fourier Shell Correlation (FSC) is frequently used as the measure of resolution, numerous concerns have been raised about this metric since it is really a measure of self-consistency and not resolution

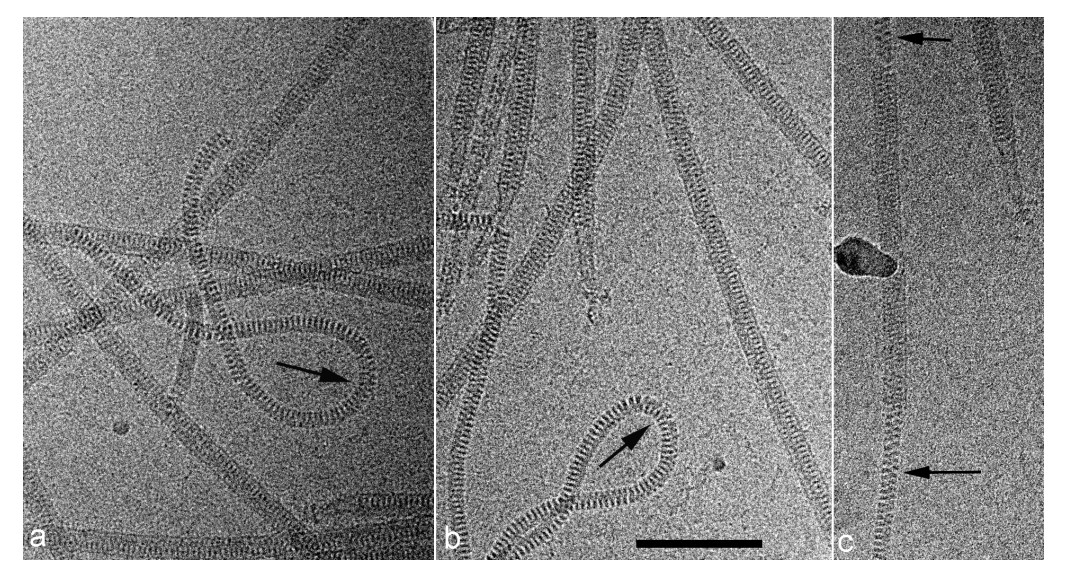

**Figure 1.** Cryo-EMs of AFV1. Arrows point to regions where the virions have been demembranated. This leads to a narrower diameter of the virions and a greatly increased flexibility, as seen by the loops in a,b. The loss of the membrane does not necessarily occur over the whole virion, as can be seen in c where a membrane-enveloped region in the center is bracketed by two regions with no membrane. The scale bar (**b**) is 1,000 Å.

(*Subramaniam et al., 2016*). Nevertheless, the 'gold standard' FSC (*Figure 4*) yields an overall resolution of 4.1 Å. We think that this is overly optimistic, and may arise from strong features in the DNA (*Figure 5*) (given the higher MW of the phosphates, the contrast is greater than for protein). A reasonable estimate (based upon comparison with the atomic model) is ~4.5 Å, but it is clear that parts of the complex (such as the outer helices facing the membrane, *Figure 3b*) are at a worse resolution, while other parts (such as the helix-turn-helix motifs on the very inside, *Figure 3c*) are at a better resolution.

The outer diameter of the virion is ~185 Å, while the diameter of the nucleoprotein core alone (excluding the membrane) is ~135 Å. Surprisingly, the membrane is only ~20–25 Å thick, compared to ~40 Å found for archaeal tetraether monolayer (*Valentine, 2007*) membranes (*Chong et al., 2003*; *Chugunov et al., 2014*) and the 40–60 Å found for other cellular membranes and the viral envelopes derived from them (*Hollinshead et al., 1999*), but a crude calculation done by integrating the cryo-EM density (which corresponds to the Coulombic potential but will be roughly proportional to mass) suggests that ~40% of the total mass of the virion is due to the membrane. The buoyant density for the AFV1 virions was previously determined using a CsCl gradient (*Bettstetter et al., 2003*), and it was consistent with other membrane-enveloped dsDNA viruses (*King et al., 2011*). The helically-averaged membrane shows two clear density peaks, with the highest one on the outside and a lower one at inner radius (*Figure 3a*). This can be seen more easily in the cylindrically-averaged density distribution of the virion, which yields the radial mass distribution (Figure 7a).

It is unlikely that any details of the membrane structure are lost due to the fact that either the helical symmetry of the nucleocapsid has been imposed upon the membrane (*Figure 3a*) or the membrane has been cylindrically averaged (Figure 7a) since the membrane is most likely a two-dimensional fluid. If there were some fine structure in the membrane (e.g., a liquid-crystalline phase with a spacing of ~5 Å) then this would be lost by the helical symmetrization (lost as well, of course, by cylindrical averaging) but would appear in averaged power spectra. Since the membrane accounts for ~40% of the mass of the virions, and since there are no features in the averaged power spectra arising from any liquid crystalline features of the membrane, all evidence suggests that it is fluid. What we have been able to observe is that if we compare the membrane with helical symmetrization with a membrane generated by cylindrically-averaging the helical density, or with that obtained from an *ab initio* cylindrical symmetry reconstruction (see Materials and methods), we see

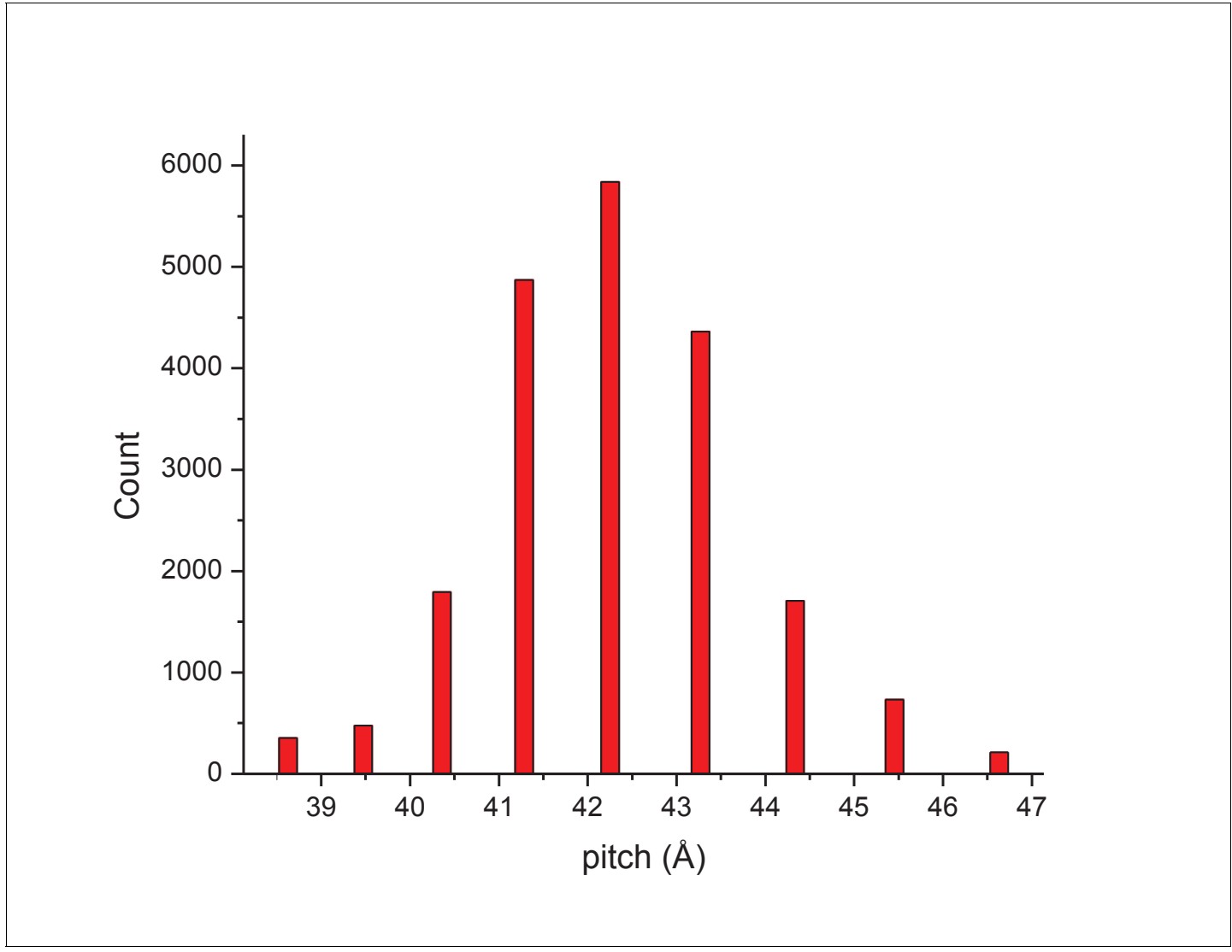

**Figure 2.** The distribution of segments sorted against references containing 1-start helices with different pitch values. The validity of this sorting was confirmed by taking power spectra from different bins, which behaved as expected and showed the helical pitch of the corresponding reference. The reconstruction was generated using segments from the central three bins.

no systematic differences. This excludes the possibility that the membrane is deformed locally by the protein in a way that the membrane would deviate from cylindrical symmetry by the presence of certain amino acid residues either facing the membrane or inserted into the membrane.

In SIRV2, the radius of the DNA is ~60 Å, while in AFV1 the supercoiling is tighter and the radius is ~30 Å. The twist of the A-form DNA in SIRV2 is 11.2 bp/turn (*DiMaio et al., 2015*) while in AFV1 it is 10.8 bp/turn: in 10 right-handed turns of the 43 Å pitch AFV1 helix, there are 93 repeats of the DNA, each with 12 bp, so there are 1116 bp per 103 (=93 + 10) right handed turns. These two values (11.2 and 10.8) bracket the 'canonical' value of 11 bp/turn frequently given for A-DNA (*Vargason et al., 2001*). Interestingly, the helical pitch in both SIRV2 and AFV1 is ~42–43 Å, but in SIRV2 there are 14.7 homodimers per turn, while in AFV1 there are only 9.3 heterodimers. It is this looser packing in AFV1 that appears responsible for the greater flexibility and disorder. It also explains why the two viruses with linear dsDNA genomes of very different sizes – 35,450 kb for SIRV2 (*Peng et al., 2001*) and 20,080 for AFV1 (*Bettstetter et al., 2003*) – have virions of approximately the same length (about 900 nm). In SIRV2 there are tight contacts across the helical turns, while in AFV1 such contacts are absent (*Figure 3b*), allowing the virions to bend, extend, and

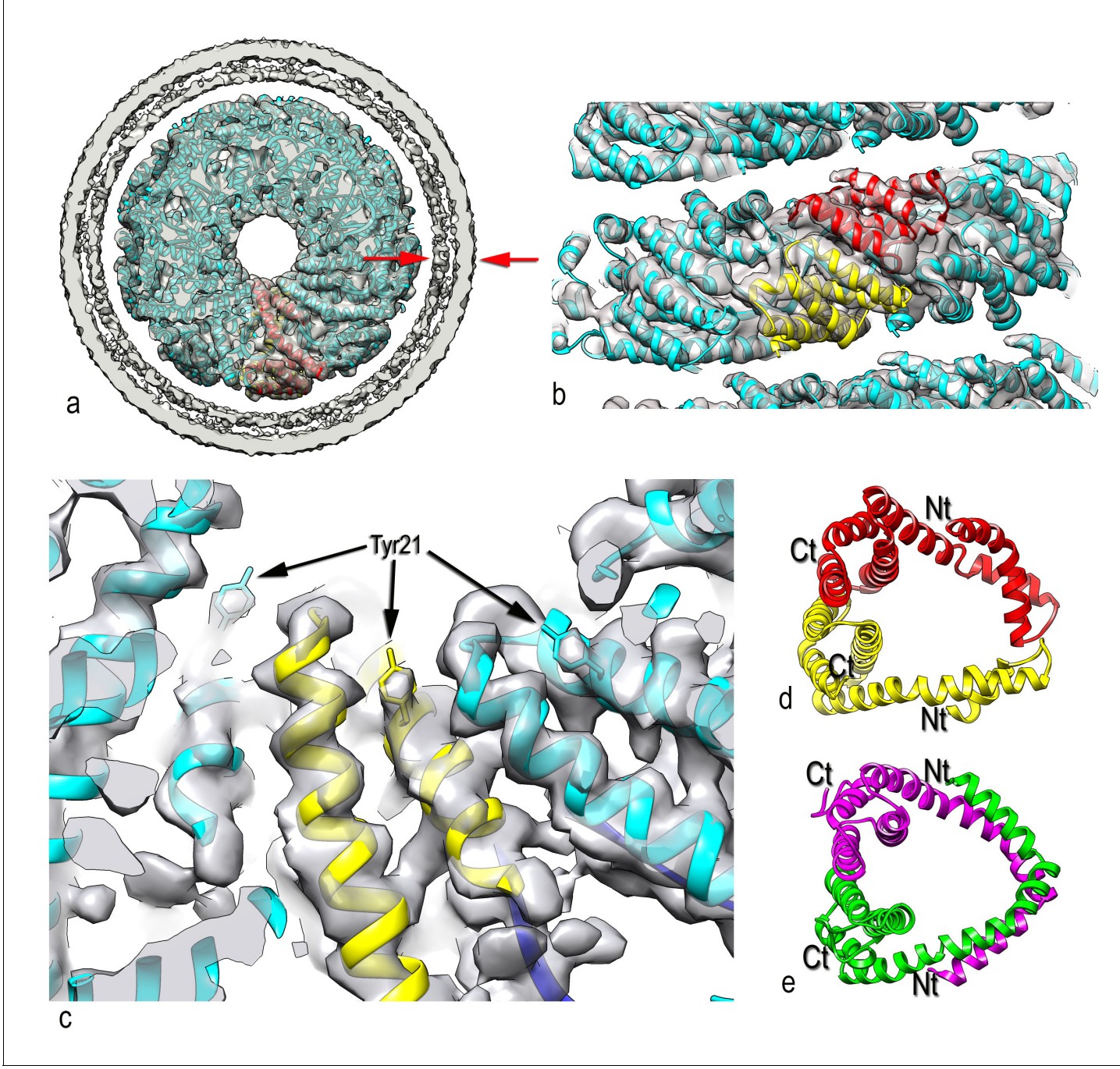

**Figure 3.** Three-dimensional reconstruction of AFV1. (**a**) A slice perpendicular to the filament axis. The red arrows define a distance of 20 Å, the approximate thickness of the membrane enveloping the virions. The membrane has a denser outer component and a less dense inner part, separated by a region of lower density. (**b**) A view of the protein core, looking from the membrane. The asymmetric unit in the virus is a pseudo-symmetric heterodimer of MCP1 (red) and MCP2 (yellow). (**c**) A view looking down the filament axis (perpendicular to that in b) with the model for the DNA phosphodiester backbone underneath the protein in blue. The helix-turn helix motif of each subunit faces into the narrow lumen. The resolution is good enough in this region that some bulky amino acids can be unambiguously located, and three Tyr21 residues are labeled. (**d**) The heterodimer in AFV1 has a pseudo-2-fold symmetry, in contrast to the homodimer in SIRV2 (**e**) which has strict 2-fold symmetry. In both, A-form DNA is bound within the central cleft. The N- and C-terminal ends in both (**d**) and (**e**) are labeled Nt and Ct, respectively.

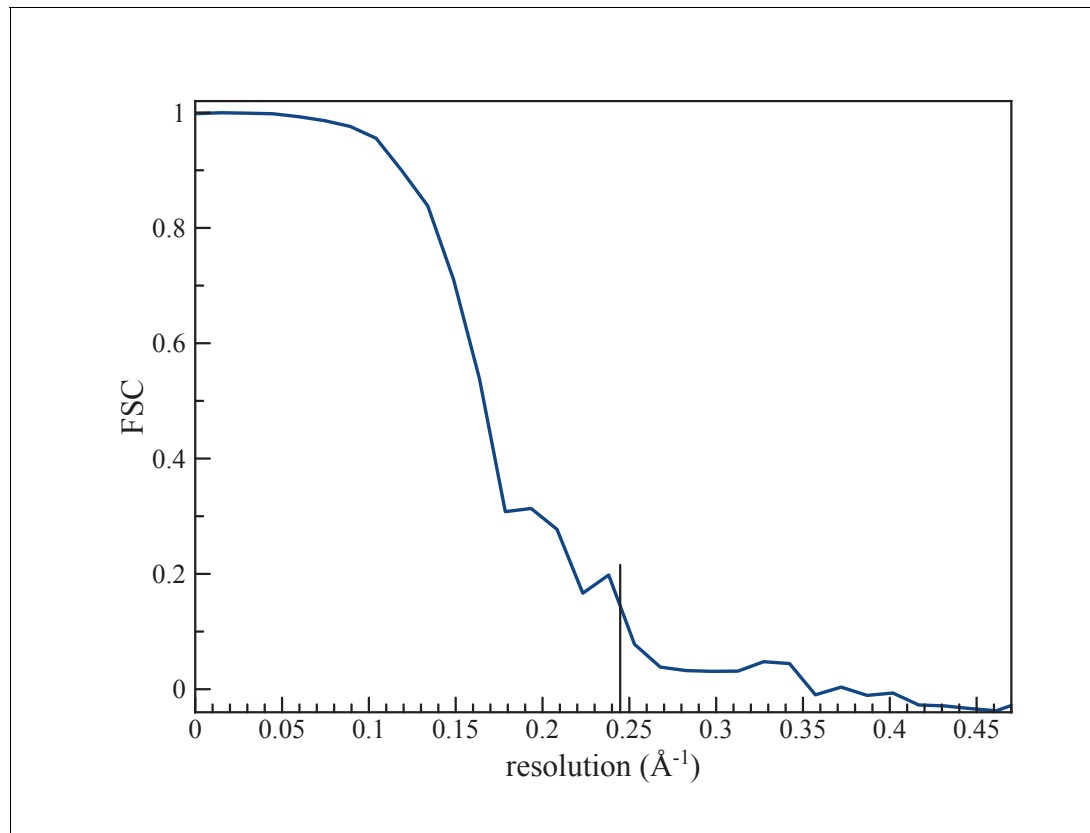

**Figure 4.** (a) An FSC curve between two reconstructions from completely independent sets of segments (having no overlap), each started independently from a full reconstruction filtered to 7 Å resolution. The FSC falls to 0.143 at 1/ (4.1 Å).

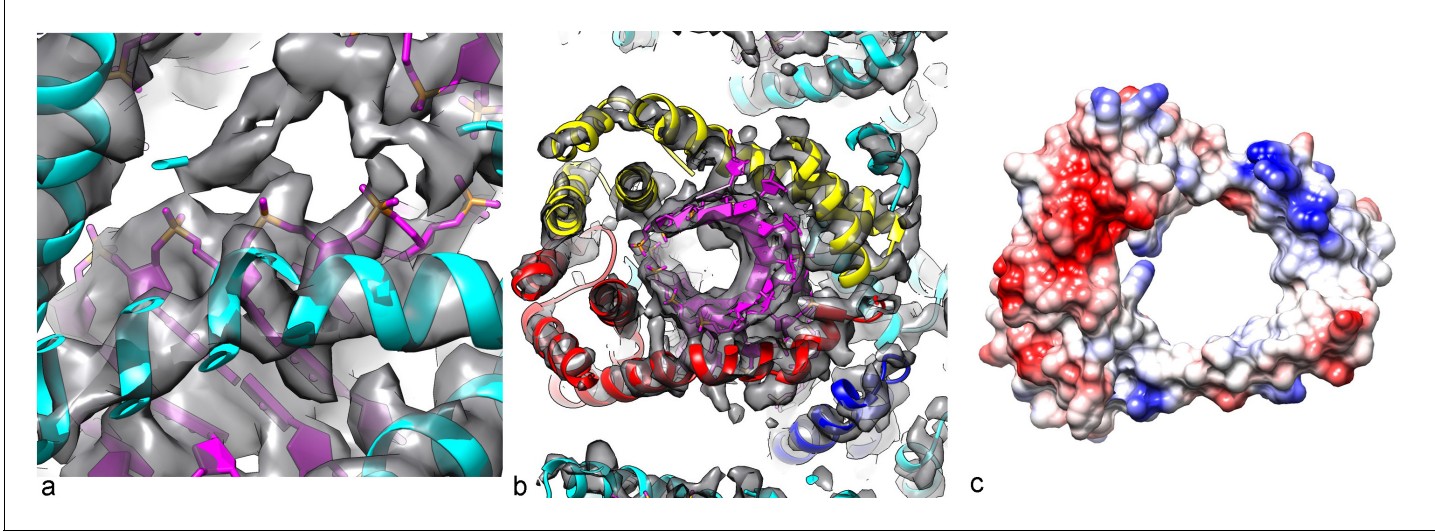

**Figure 5.** Packing of the DNA in the virion. (a) The phosphate backbone of the DNA model can be fit nicely into the density map, as most of the phosphate groups are well resolved. (b) A slice perpendicular to the DNA axis through the map and model. As expected for A-form DNA (and in contrast to B-form), a hole is seen along the DNA axis, with the bases surrounding this cavity. (c) The Coulombic potential for the AFV1 capsid heterodimer shows significant positive regions (blue) surrounding the DNA, with negative regions (red) away from the DNA.

compress. At the same time, due to looser protein packing, the AFV1 genome is not completely covered by the protein, while it is in SIRV2. Consequently, the lipid envelope provides a necessary protection to the AFV1 genome in the highly acidic environment of the natural habitat, rationalizing the presence of the envelope in *lipothrixviruses*. When the membrane is removed (we assume as an artifact of specimen preparation) the virions become much more flexible (*Figure 1*). Since the membrane, which has fluid-like properties, is unlikely to be directly responsible for the increased rigidity of the enveloped virions, it suggests that the presence of the membrane constrains the protein and thus indirectly imparts rigidity to the structure.

It was originally proposed from a crystallographic study that the two capsid proteins MCP1 and MCP2 would be packed very differently in the virion (*Goulet et al., 2009*). A model, based upon crystal structures of most of one capsid protein and a fragment of the second one, proposed that one of the capsid proteins formed an inside core of the virion, with DNA wrapping around it, while the other subunit was on the outside of the DNA and partially inserted into the membrane. Surprisingly, we find that the two capsid proteins form a pseudo-symmetric heterodimer (*Figure 3d*) that resembles in many ways the symmetric homodimer found in SIRV2 (*Figure 3e*), and that both interact with the DNA in an equivalent manner. We have accounted for all of the amino acids in the two capsid proteins with the exception of 5 N-terminal residues in both MCP1 and MCP2. However, these residues would be too far from the membrane to contact it. Further, we see no density extending from the protein to the membrane.

There are two main differences between the AFV1 and SIRV2 dimers: (1) In SIRV2 the N-terminal tail forms a long helix with a kink that allows it to continuously wrap around the DNA (*Figure 3e*), while in AFV1 the N-terminal region of both MCP1 and MCP2 form helix-turn-helix motifs which fold back to cover the DNA on both sides (*Figure 3d*); (2) In SIRV2 the 2-fold axis of the dimer is perpendicular to the helical axis (and goes through the 2-fold axis in the DNA), generating an overall bipolar symmetry for the virion, while in AFV1 the pseudo-2-fold axis of the heterodimer is tilted by 25.7° and does not intersect the helical axis, so that the virion has an overall polarity visible at fairly low resolution. Details of the wrapping of the A-form DNA by the heterodimer are shown in *Figure 5*, where it can be seen that a positive Coulombic potential would surround the negatively-charged phosphate backbones of the DNA.

The fact that the membrane is only 20–25 Å thick, half of regular lipid membranes, has led us to investigate the membrane further. Since the membrane lipids would not be synthesized by the virus but must come from the host, we first compared the distribution of lipids (*Figure 6a*) found in the host with those found in the virion membrane (*Figure 6b*). There is a striking difference in the distributions showing that the incorporation of the glycerol dibiphytanyl glycerol tetraether (GDGT) lipids from the host is highly selective. While the single most dominant species in the host is GDGT-4 (containing 4 cyclopentane moieties), in the virion it is GDGT-0 (containing no cyclopentane moieties), found as only a few percent of the total host membrane lipids. Nevertheless, GDGT-0 is actually one of the most common archaeal membrane lipids (*Schouten et al., 2013*; *Villanueva et al., 2014*). Furthermore, it is generally the dominant, or one of the dominant, archaeal lipids in environmental samples taken from soils, rivers, lakes and oceans accounting for >40% of all GDGTs detected (*Schouten et al., 2013*). The selective incorporation of host lipids in a viral membrane has previously been described, for example, in influenza budding from mammalian cells (*Gerl et al., 2012*), or in a virus budding from algae (*Maat et al., 2014*). Such selective incorporation could be driven by direct lipid binding by capsid proteins, enrichment of certain lipid species at sites of viral budding, or physical properties of the viral envelope that cause partitioning of lipids into or out of the nascent envelope during viral budding. Because GDGT-0 is more flexible than the cyclopentane-containing GDGT lipids (*Schwarzmann et al., 2015*), it can better adopt the horseshoe conformations that have a lower free energy in the highly curved AFV-1 envelope (*Galimzyanov et al., 2016*). We therefore hypothesize that selective partitioning of GDGT lipids due to the curvature of the envelope is the mechanism for GDGT-0 enrichment in the AFV-1 membrane.

Knowing the lipid composition of the virion, we used molecular dynamics (MD) to model the viral membrane. Multiple simulations were performed of GDGT-0 lipids arranged cylindrically around the capsid assembly in different densities and orientations. Lipids were modeled with a single phosphoinositol headgroup, as this is the smallest headgroup commonly found on GDGT lipids in the *A. hospitalis* host (the others are dihexose and sulfonated trihexose headgroups). These lipids frequently adopted a U-shaped or horseshoe conformation, and these horseshoe-rich envelopes with a mix of

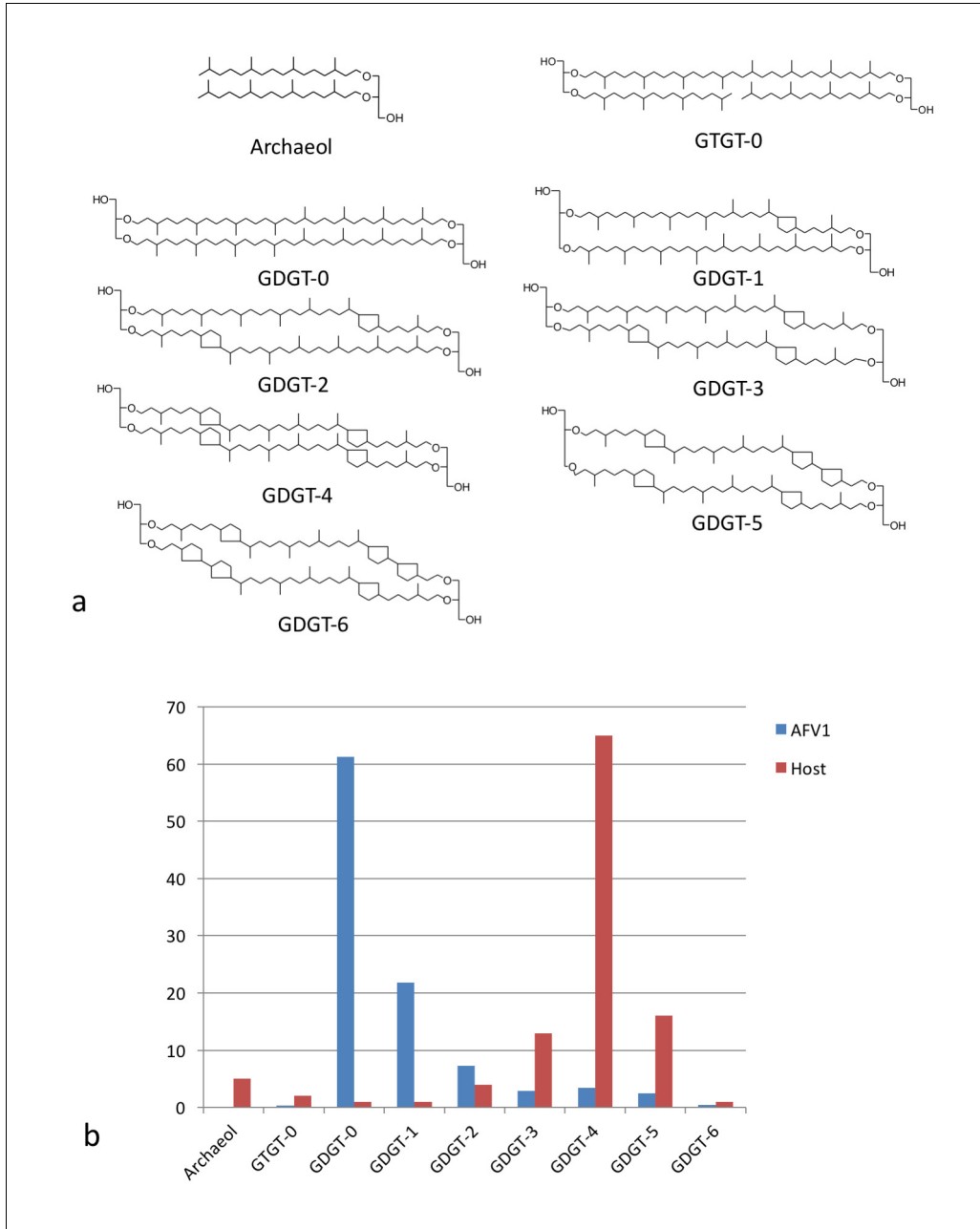

**Figure 6.** Lipid distribution of virions different from host cells. (**a**) Chemical diagrams for the lipids found in *Acidianus hospitalis* and AFV1. (**b**) The distribution of lipids in the host membrane (red) differs significantly from the distribution found for AFV1 (blue). The scale is in percentage.

'inward-facing' and 'outward-facing' lipids were the only ones that stably maintained the curvature and thickness observed in the radial density profile from cryo-EM. These lipids still form a monolayer one lipid thick, but the lipids in the monolayer have a mixed orientation. A horseshoe lipid conformation from simulation was therefore used to fit the cryo-EM radial density profile; the best-fit arrangement features 40% of lipids with headgroups facing inwards towards the nucleocapsid and 60% of lipids with headgroups facing towards the outside (*Figure 7*). Structural models simulated with this lipid orientation maintained a stable envelope structure with the thickness, curvature, propensity to horseshoe conformations, and the slight ~8 Å water-filled gap between envelope and capsid observed by cryo-EM, similar to a surface-supported membrane. The density in this gap observed

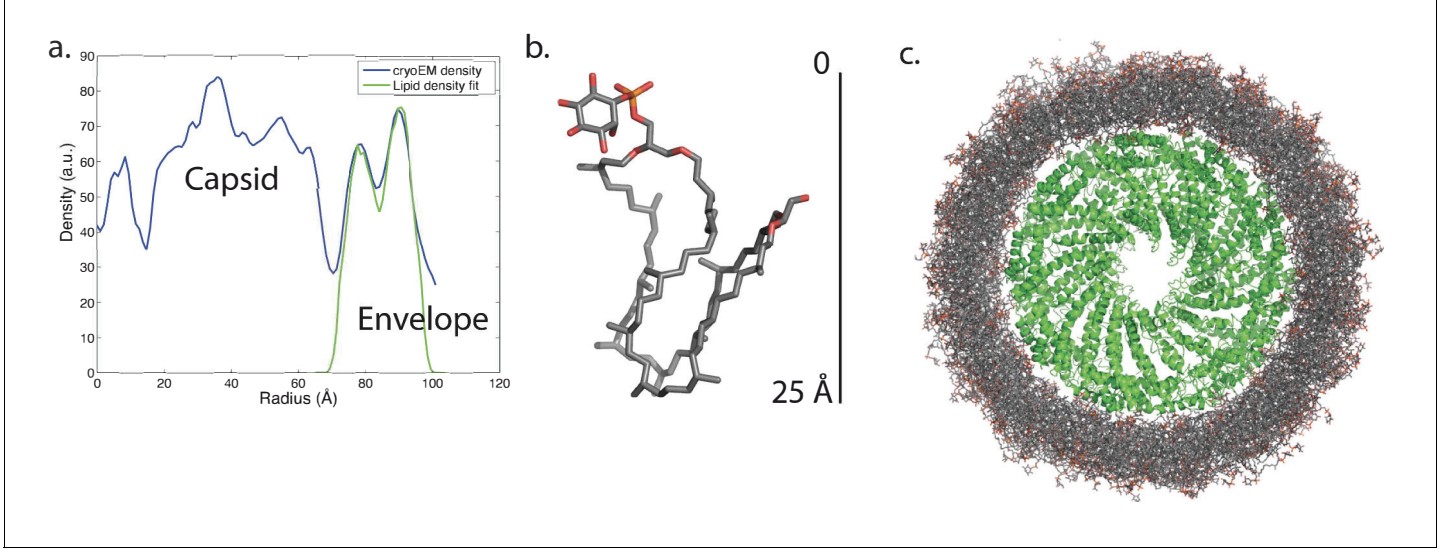

**Figure 7.** Modeling the viral membrane. (**a**) The cylindrically averaged density profile from EM (blue curve) is well fit by a cylindrical envelope (green curve) of phosphoinositol-GDGT0 lipids in horseshoe conformations (**b**) with 60% having headgroups facing away from the capsid and 40% having headgroups facing towards the capsid. Molecular dynamics simulations of the protein capsid and phosphoinositol-GDGT0 lipids constructed in this arrangement produced a stable envelope rich in horseshoe-conformation lipids (**c**), while all other envelope arrangements tested failed to maintain the experimentally-derived thickness of 20–25 Å. The density peak at ~30 Å radius (**a**) arises from the DNA. The central cryo-EM density (radius <15 Å) could not be explained by the capsid proteins, and most likely involves either a minor viral protein or a host protein (**Figure 7—figure supplement 1**). Since the symmetry of the virion was imposed on this density, which likely does not have such a symmetry, the density was uninterpretable and removed from the other figures. Reconstructed density profiles from the simulations are shown in **Figure 7—figure supplement 2**, accompanied by movies of 20 envelope lipids in **Video 1** and of the entire envelope and interfacial water layers in **Video 2**.

The following figure supplements are available for figure 7:

**Figure supplement 1.** A central disconnected density (yellow arrow) cannot be explained by the two capsid proteins and most likely involves either a minor viral protein or a host protein.

**Figure supplement 2.** Computed radial density from molecular dynamics simulations.

by cryo-EM was the same as the solvent outside the virus, further suggesting that the region between the envelope and the polar capsid surface and the envelope is similar to that between a supported lipid bilayer and its planar support (**Ajo-Franklin et al., 2005**; **Koenig et al., 1996**).

Simulations very robustly reproduced the width and placement of the envelope density compared to the cryo-EM data, and the density was stable over the course of multiple independent simulations. However, the lipids in the simulations were somewhat more disordered than suggested by the cryo-EM density, such that the double-peak density profile from cryo-EM was smoothed into a broader single peak, and most but not all simulated lipids were in horseshoe conformation. This could result from one of three factors: (1) a slight mismatch in the estimated density of lipids in the viral envelope

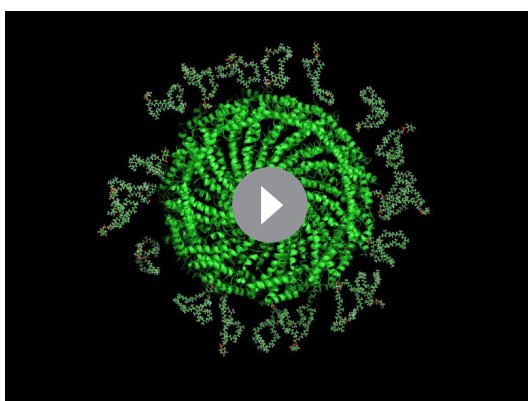

**Video 1.** Conformational dynamics of 20 lipids from the simulated AFV1 envelope. Rendered in this movie are 20 sequential lipids from the simulated AFV1 envelope in stick form and the capsid protein in cartoon form. Frames are at nanosecond intervals.

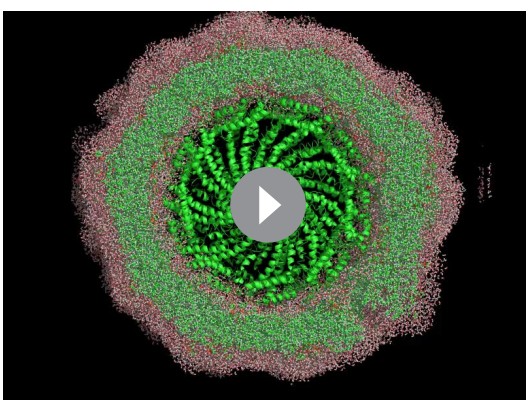

**Video 2.** Conformational dynamics of AFV1 envelope lipids and interfacial water. Rendered in this movie are the simulated AFV1 envelope in stick form, all water molecules within 10 Å, and the capsid protein in cartoon form. Frames are at nanosecond intervals.

leading to lateral pressure stresses in the envelope, (2) a larger lipid headgroup present in AFV-1 envelopes than those used in simulations—the simulations used a phosphoinositol headgroup and glycerol backbone as the minimal headgroups found on host lipids, but larger headgroups are also possible, or (3) factors internal to the simulation such as insufficient sampling time or slight mismatches in lipid parameterization. Despite this minor disordering of the lipid tails, the simulated AFV-1 envelope stably maintained a thickness consistent with a single horseshoe-conformation lipid with a thin layer of water between the capsid and envelope. Control simulations that used either incorrect lipid density or single-orientation lipids rather than an 'in/out mix' of headgroup orientations did not maintain these features over equivalent simulation timescales. These findings are thus robust and highly consistent with the experimental data.

The simulation models therefore suggest that a mixture of inward-facing and outward-facing lipids primarily in horseshoe conformations is physically stable surrounding a highly curved polar capsid. These simulations match the thickness of the envelope in the electron-density profile and well explain the gap between capsid and envelope as a water layer, but they are not sufficiently powered to distinguish between some-horseshoe and all-horseshoe conformational distributions due to slow conformational relaxations of the lipids and initial-value sensitivity. We have tested sufficient initial conditions to say with confidence that (1) a canonical 'straight' tetraether lipid conformation is not compatible with the cylindrical curvature of the capsid; (2) an in/out orientational mix is necessary to capture the gap in density between the capsid and the envelope; and (3) multiple starting conditions with in/out horseshoe start states all produce a stable envelope with a thickness matching that observed experimentally.

These simulations thus provide a specific structural model for the lipids to fit our experimental findings that GDGT-0 lipids in the envelope must occupy a horseshoe conformation based on the cryo-EM density profile of the envelope. The models also predicted that the membrane would account for ~43% of the total mass of the virion, in excellent agreement with the ~40% estimate from the cryo-EM density integration.

## Discussion

AFV-1 is striking because its envelope differs substantially in composition and structure from those of previously described viruses. Because we have combined multiple experimental approaches with computational modeling to analyze the AFV-1 envelope, we briefly recapitulate the lines of evidence for each major finding before going on to discuss some of the important implications of this previously unappreciated envelope structure. Our cryo-EM density data show an envelope surrounding the viral capsid that is ~20 Å in thickness; our mass spectrometry data show that this envelope is composed predominantly of lipids with a GDGT-0 core. The only known conformation of GDGT-0 lipids to form a layer that is <40 Å in thickness is the horseshoe conformation, which has been characterized previously at fluid-air interfaces. Our cryo-EM density data further show an 8 Å region with density corresponding to water between the protein capsid and the lipid, and the structural refinement of the capsid yields a hydrophilic surface. Our computational models then explain these findings via a horseshoe-conformation monolayer with mixed orientation such that (1) GDGT-0 acyl tails are not fully exposed to the external solvent or the polar capsid and (2) the tight curvature of the capsid is well matched by the envelope (which has a radius of curvature on the inside of the membrane of ~70 Å). Thus, the horseshoe-conformation lipid envelope is strongly supported by the

experimental data themselves, while the computational model specifies the likely orientation of lipids within this envelope and provides a detailed model of molecular structure.

Despite extensive exploitation of archaeal tetraether lipids for therapeutic purposes, such as archaeosome-based delivery of drugs, cancer vaccines, antigens, genes, etc. (*Kaur et al., 2016*) remarkably little is known about the actual structure of membranes in *Acidianus* specifically and *Sulfolobales* in general. The studies on membranes of *Sulfolobales* have thus far largely focused on lipid composition in different organisms and on investigation of lipid mixtures in in vitro systems. The lipid composition of *Acidianus hospitalis*, which we report in this study, is very similar to that previously determined in a related organism, *Sulfolobus solfataricus* (*Quemin et al., 2015*) which has a ~5 nm-thick membrane. Interestingly, it has been shown that a spindle-shaped virus SSV1 released from *Sulfolobus* cells by budding, similar to AFV1, has a membrane enriched in GDGT-0 which is also considerably thinner compared to the host membrane (*Quemin et al., 2016*). However, this observation remained unexplained. It thus appears that the lipid conformation described in our current study might be more general in enveloped viruses of archaea.

A number of in vitro studies have described archaeal lipids forming U-shaped structures at an air-water interface (*Gliozzi et al., 1994*; *Köhler et al., 2006*; *Melikyan et al., 1991*; *Patwardhan and Thompson, 2000*; *Tomoaia-Cotisel et al., 1992*). Since air is extremely hydrophobic, the acyl chains face the air while the polar headgroups of these lipids face the water. The presence of cyclopentane rings (found in the main host species, GDGT-4) has been suggested to rigidify the lipids, making them unable to form a horseshoe (*Gliozzi et al., 1994*), in agreement with another study which found that more rigid tetraether lipids could not form a horseshoe at the air-water interface while the more flexible lipids did (*Patwardhan and Thompson, 2000*). Furthermore, theoretical studies suggest that membranes formed from horseshoe conformation lipids have lower curvature energies (*Galimzyanov et al., 2016*) and would thus be energetically favored on the highly curved AFV-1 surface. This may be the main driving force for the exquisite selectivity seen for the incorporation of host lipids in the viral membrane. A biological role for such a horseshoe conformation has not previously been suggested or found. Our study demonstrates that besides the canonical bacterial/eukaryotic membrane bilayer and archaeal monolayer, there is a third type of biological membrane, the viral horse-shoe membrane layer.

The observation that a membrane that envelops a virus can be formed from lipids in such a conformation opens the door to designing such membranes for applications from drug delivery to nanotechnology. Since the archaeal lipids have been shown to resist phospholipases, extremes of temperature and pH, and can even survive autoclaving, the membrane described here has many potential applications (*Patel and Sprott, 1999*).

## Materials and methods

### Sample preparation, Cryo-EM and Image Analysis

The virus AFV1 was purified for electron microscopy as described earlier (*Bettstetter et al., 2003*); for lipid analysis additional purification was performed on a sucrose gradient (*Quemin et al., 2015*). The purified virus preparation (3 μL, 1–2 μg/μl) was applied to lacey carbon grids that were plasma cleaned (Gatan Solarus) and vitrified in a Vitrobot Mark IV (FEI, Inc.). Grids were imaged in a Titan Krios at 300 keV, and recorded with a Falcon II direct electron detector at 1.05 Å per pixel, with seven 'chunks' per image. Each chunk, containing multiple frames, represented a dose of ~20 electrons/Å$^2$. A total of 557 images (each 4 k x 4 k) were selected that were free from drift or astigmatism, and had a defocus less than 3.0 μm. The program CTFFIND3 (*Mindell and Grigorieff, 2003*) was used for determining the Contrast Transfer Function (CTF) and the range used was from 0.6 to 3.0 μm. The SPIDER software package (*Frank et al., 1996*) was used for most subsequent steps. The CTF was corrected by multiplying each image by the theoretical CTF, both reversing phases where they need to be reversed and improving the Signal-to-Noise ratio. This multiplication of the images by the CTF is actually a Wiener filter in the limit of a very poor SNR. The program e2helixboxer within EMAN2 (*Tang et al., 2007*) was used for boxing long filaments from the micrographs, and 546 such boxes of varying length were extracted. Overlapping boxes, 384 px long with an 8 px shift between adjacent boxes (~twice the axial rise per subunit) were extracted from these long filaments, yielding 215,549 segments that were padded to 384 × 384 px. The CTF determination and particle

picking came from the integrated images (all seven chunks), while the segments used for the initial alignments and reconstruction came from the first two chunks.

The determination of the helical symmetry was by trial and error, searching for a symmetry which yielded recognizable secondary structure (*Egelman, 2014*). The IHRSR algorithm (*Egelman, 2000*) was used for the helical reconstructions, starting from a solid cylinder as an initial model. Once the correct symmetry was determined (an axial rise of 4.6 Å and a rotation of 38.7° per subunit) it was apparent that the pitch was quite variable, and segments were sorted using references that had a continuous 1-start helix with different pitch. Segments were excluded if they corresponded to a pitch less than or equal to 40.5 Å or greater than or equal to 44.1 Å (*Figure 2*), reducing the number of segments to 119,495. The final reconstruction was generated by imposing the helical parameters found for each segment using the first two chunks on segments containing only the first chunk (~20 electrons/Å$^2$) and using these for the back-projection in SPIDER. The variability in the structure was further overcome by only symmetrizing the central third (128 px) of the 384 px long asymmetric reconstruction.

Since the images had been multiplied by the CTF twice (once by the microscope and once by us in phase correction), the amplitudes of the final volume were divided by the sum of the squared CTFs. The reconstructed volume (which has a very high SNR from the extensive averaging) is corrected only in Fourier amplitudes by dividing by the sum of the squared CTFs. This is a Wiener filter in the limit of a very high SNR. The map was also sharpened using a negative B-factor of 220.

Resolution was estimated by dividing the data set into two independent halves, such that there was no overlap in segments between one set and the other. These were used to iteratively generate (after 20 cycles) two reconstructions for a Fourier Shell Correlation (FSC), starting with a reference volume for each set that was filtered to 7 Å. Choosing the FSC = 0.143 threshold yielded a resolution of 4.1 Å (*Figure 4*).

The radial density profile was generated by cylindrically averaging the reconstructed volume, since the cylindrically-averaged radial density distribution after helical symmetrization is actually the same as the unsymmetrized mean radial density distribution. Cylindrically-symmetrizing the IHRSR helical reconstruction (*Figure 7a*) leaves only the equatorial terms in a Fourier-Bessel synthesis. That is the same as the approach of reconstructing by assuming cylindrical symmetry, which is the J0 Fourier-Bessel transform of the equator using Fourier methods. We have done this in real space using ~9000 segments classified as having an out-of-plane tilt of <0.5°, and treated these as cylindrically-symmetrical objects, reconstructed them using standard back-projection methods and then corrected for the CTF. The density profile for the membrane is the same as in *Figure 7a*.

A small central density within the lumen of the reconstruction (*Figure 7—figure supplement 1*) with no connectivity to the nucleocapsid could not be explained by the capsid proteins, and most likely involves either a minor viral protein or a host protein. Since the symmetry of the virion was imposed on this density, which likely does not have such a symmetry, the density was uninterpretable and removed from the other figures.

## Model-building with Rosetta

Models were initially constructed using RosettaCM (*Song et al., 2013*). As template structures we considered the cryoEM structure of SIRV (PDB id: 3J9X) as well as the crystal structures of the two components (PDB ids: 3FBZ and 3FBL). Sequence alignments for both molecules in the asymmetric unit were generated with hhsearch (*Söding et al., 2005*). Threaded models were then docked into density. Initially, visual inspection was used to assign each of the two components into the asymmetric unit. Modelling was carried out in the context of the entire assembly, using the helical parameters determined from map construction. Using RosettaCM, a total of 5000 models were generated, of which the top 10% were chosen by energy and the top model was chosen by agreement to density.

Following the procedures from SIRV (*DiMaio et al., 2015*), we also built models for the A-form DNA present. Initially, models with 11 and 12 base pairs of DNA per repeat were docked into the density map. Then, these models were randomly perturbed and minimized in Cartesian space with Rosetta, to correct the geometry at the junction between asymmetric units, as well as to better improve the local fit of model to density. As with SIRV, a fit of 12 base pairs per repeat showed the best agreement with the data, with the top five models showing very good convergence on a single conformation (*Figure 8a*). An additional refinement step in RosettaCM modeled the protein and

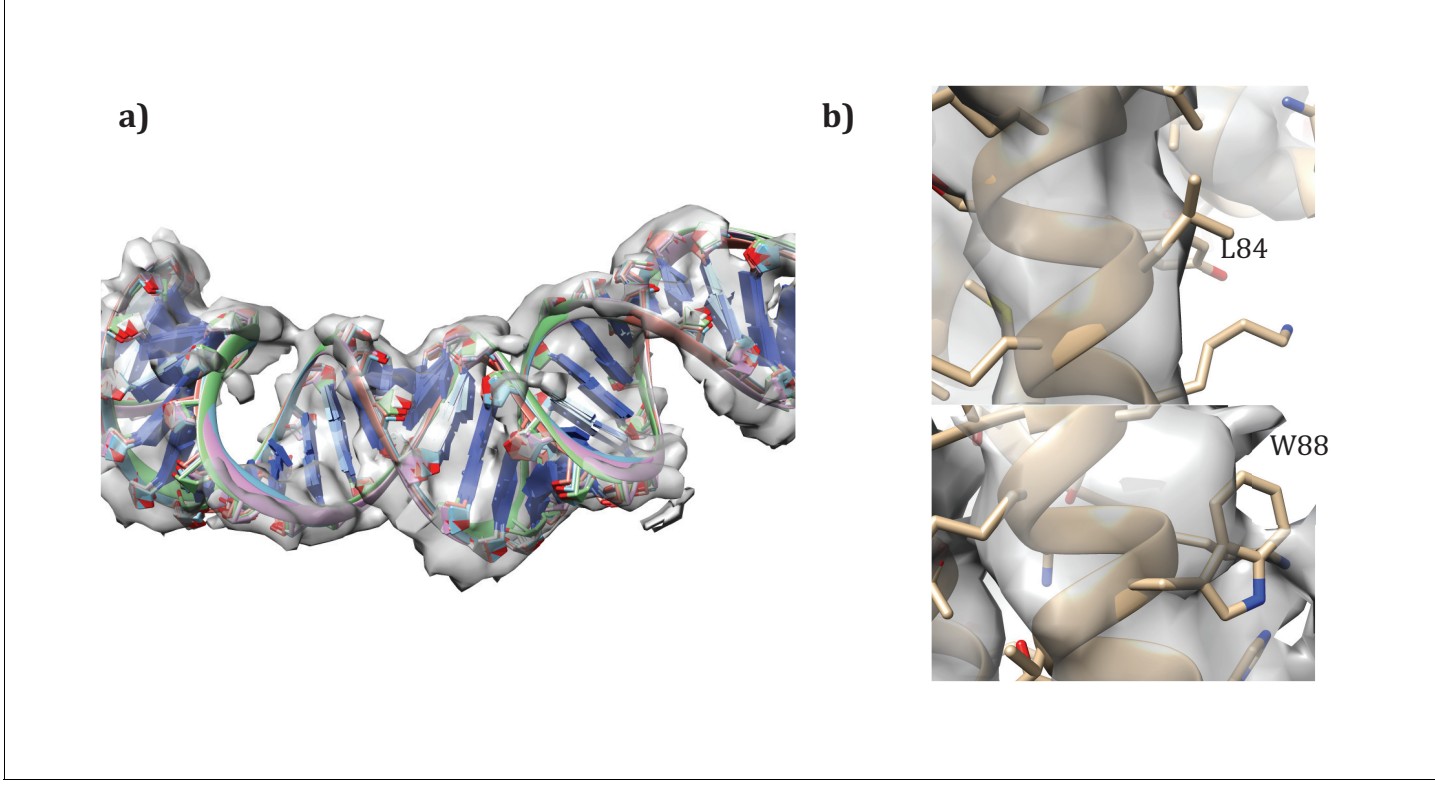

**Figure 8.** Details of the modeling. (a) Convergence of DNA model refinement results. Sampling DNA conformations revealed a very tight cluster of models with 12 bp/repeat, showing good agreement with the density. (b) Two corresponding residues that helped guide initial placement of the two distinct subunits: the extra density visible around W88 compared to the corresponding residue, L84, where no sidechain density is visible.

DNA simultaneously; for this step the DNA was held fixed and only protein conformation was optimized.

## Validation of molecule assignment

Unlike SIRV, the asymmetric unit contains two different molecules. While the assignment was first visually assigned, using the density of aromatic residues – particularly W88 – to place the subunits (*Figure 8b*), we wanted to quantify the degree of fit of the two. Therefore, we set up an additional RosettaCM run where the order of chains was reversed. Such modeling gave a clear signal for our initial assignment, with an integrated FSC (10–4.8 Å) of 0.532 for our original assignment versus 0.495 for the flipped assignment, and a Rosetta energy of −11534 for the original assignment versus −11467 for the flipped assignment. Following these experiments we were confident in our original assignment of subunits to density.

The presence of an outer membrane in AFV1 suggested a possible conformational change where hydrophobic residues in the outermost helices faced outward. In order to precisely identify this conformational change, we sampled all possible outer helix registry shifts by ±4 residues for each of the 4 helices, giving a total of $9^4$ = 6561 possible starting models. Each of these starting models was refined against the density and assessed by fit to the data in high-resolution shells (integrated FSC from 10 to 4.8 Å). A total of 5 models were generated from each starting point. The best 10 models by the FSC were then subject to additional rounds of refinement, where the connecting loops were rebuilt in RosettaCM, followed by all-atom refinement. Rescoring these full-length models again gave a strong signal for the top placement from the original 6561 placements.

## Lipid analysis

The freeze-dried host-cell preparation and the virion preparation were directly acid hydrolyzed by refluxing with 5% HCl in methanol for 3 hr following Pitcher et al. (*Pitcher et al., 2011*) to release GDGT lipids. GDGT lipids were analyzed by high-performance liquid chromatography/atmospheric pressure chemical ionization-mass spectrometry (*Hopmans et al., 2016*). The mass spectrometer was operated in single ion mode (SIM) to monitor GDGTs with 0 to 8 cyclopentane moieties. Relative abundances of GDGTs were determined by integrating peak areas of the SIM signal.

## Modeling the membrane

Simulations were performed using the GROMACS molecular dynamics package (*Pronk et al., 2013*) using the CHARMM36 parameter set (*Best et al., 2012*; *Klauda et al., 2010*) modified to include GDGT-0 lipids following the example of previous work on tetraether lipids (*Shinoda et al., 2004*). In the absence of definitive mass spectrometry data on the headgroups of the GDGTs in the viral envelope, phosphoinositol was selected as a common archaeal tetraether lipid headgroup. To construct models of the AFV1 envelope, lipids were placed in a cylindrically symmetric arrangement around a capsid cylinder 10 nm in length. The lipid lateral density, fraction inward versus outward facing, and radius of the envelope cylinder were all systematically varied in initial system setup, and the resulting simulations were assessed with regard to physical stability and maintenance of the capsid thickness observed on cryo-EM. After lipid placement, the system was energy-minimized and then placed in a $26 \times 26 \times 10$ nm water box with 150 mM NaCl and sufficient counterions to neutralize the system. Molecular dynamics simulations were run using 1 fs time step with a 1.2 nm cutoff, with long-range electrostatics calculated using Particle Mesh Ewald (*Darden et al., 1993*). Simulations were also performed with and without positional restraints of 1000 kJ mol$^{-1}$ nm$^{-2}$ on the capsid protein non-hydrogen atoms to enforce the cryoEM-refined structure. The capsid maintained both its structure and agreement with the EM radial density distribution well even in the absence of such restraints. Temperature was maintained at 310K using the velocity rescaling thermostat (*Bussi et al., 2007*); simulations were also run at 277K to approximate storage conditions and 353K to approximate physiological temperature with no gross structural changes, as anticipated from previous studies of archaeal tetraether lipids (*Koga and Morii, 2005*). A total of 30 such simulations were run, with envelope arrangements that were stable and consistent with cryo-EM-derived widths continued >60 ns each. Density profiles are shown for envelope models containing ~500 phosphoinositol-GDGT-0 lipids in a 60:40 inward:outward facing arrangement around a 100 Å cylindrical section of the capsid protein and containing a total of approximately 700,000 atoms including solvent and ions.

## Data deposition

The cryo-EM map was deposited in the EMDB with accession number EMD-8780, and the atomic model was deposited in the PDB with accession number 5W7G.

# Acknowledgements

This work was supported by NIH GM035269 (to EHE) and GM098304 (to PK), and by Agence Nationale de la Recherche grant ANR-13-BSV3-0017-01 (to DP) Computational resources were supported by Google. We thank Denise Dorhout (NIOZ) for analytical assistance.

# Additional information

### Competing interests

EHE: Reviewing editor, *eLife*. The other authors declare that no competing interests exist.

### Funding

| Funder | Grant reference number | Author |
|---|---|---|
| National Institutes of Health | GM035269 | Edward H Egelman |
| Agence Nationale de la Recherche | ANR-13-BSV3-0017-01 | David Prangishvili |

The funders had no role in study design, data collection and interpretation, or the decision to submit the work for publication.

## Author contributions

PK, Conceptualization, Software, Supervision, Funding acquisition, Investigation, Writing—original draft, Project administration, Writing—review and editing; FD, Software, Investigation, Visualization, Writing—original draft, Writing—review and editing; XY, Software, Formal analysis, Investigation; SL-S, MK, Investigation; SS, DP, Investigation, Writing—original draft, Writing—review and editing; EHE, Conceptualization, Funding acquisition, Investigation, Writing—original draft, Project administration, Writing—review and editing

## Author ORCIDs

Peter Kasson, http://orcid.org/0000-0002-3111-8103
Frank DiMaio, http://orcid.org/0000-0002-7524-8938
Mart Krupovic, http://orcid.org/0000-0001-5486-0098
Edward H Egelman, http://orcid.org/0000-0003-4844-5212

# Additional files

## Major datasets

The following datasets were generated:

| Author(s) | Year | Dataset title | Dataset URL | Database, license, and accessibility information |
|---|---|---|---|---|
| Frank DiMaio, Xiong Yu, Soizick Lucas-Staat, Mart Krupovic, Stefan Schouten, David Prangishvili, Edward H Egelman | 2017 | Cryo-EM map | https://www.ebi.ac.uk/pdbe/entry/emdb/EMD-8780 | Publicly available at the EBI European Nucleotide Archive (accession no: EMD-8780) |
| Frank DiMaio, Xiong Yu, Soizick Lucas-Staat, Mart Krupovic, Stefan Schouten, David Prangishvili, Edward H Egelman | 2017 | Atomic Model | http://www.rcsb.org/pdb/search/structidSearch.do?structureId=5W7G | Publicly available at the RCSB Protein Databank (accession no: 5W7G) |

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
