## [Decision Letter]

Thank you for submitting your article "An envelope of a filamentous hyperthermophilic viruscarries lipids in a horseshoe conformation" for consideration by *eLife*. Your article has been reviewed by three peer reviewers, and the evaluation has been overseen by Sriram Subramaniam as the Reviewing Editor and Richard Aldrich as the Senior Editor. The following individual involved in review of your submission has agreed to reveal his identity: Huilin Li (Reviewer #3).

The reviewers have discussed the reviews with one another and the Reviewing Editor has drafted this decision to help you prepare a revised submission.

Summary:

In this manuscript, Kasson et al. report a modeled structure for the membrane of a filamentous, flexible archaeal virus, AFV1. The authors pay particular attention to the structure of the lipid envelope formed from tetraether lipids that have, in effect, a headgroup at each end of a pair of alkyl chains. The authors identify the main species of the viral lipids as GDGT0, which is more flexible than the GDGT4, the main component of the host cell membrane. Based on the finding from the cryo-EM studies that the membrane is thinner than usual, the authors present computer modeling simulations to suggest that the lipid GDGT0 may take on a hairpin-like folded configuration rather than an extended conformation.

Essential revisions:

There was overall agreement of the novelty of the finding, but significant concerns emerged both about the computational aspects of the study as well as the style of presentation of the results.

1) The lack of sufficient detail about the MD simulations was seen as a major weakness of the manuscript, and led to the concern that something may be amiss in the modeling. Reviewers would need to see snapshots of the 'bilayer' and time-averaged transbilayer distributions of the key molecular features of the lipids discussed in the paper as well as a movie that shows the motions of the lipids and associated waters to be convinced.

2) Reviewers were concerned that there was not a direct measurement of the lipid content, for example by measuring the buoyant density on a suitable gradient known to provide accurate results for other kinds of viruses.

3) It is important to provide a better clarification of the nomenclature and concepts involved in characterizing the membrane, differentiating the "monolayer" vs. "bilayer" aspects of the membrane, including a better representation in figures (e.g. close-up images from Figure 7).

4) Title overstates the results, and should reflect that the proposed membrane architecture is derived from a computational model.

5) The reviewers expressed concern at the long and discursive nature of the manuscript in several places, instead of keeping the text focused on the main results.

6) One reviewer had the following analysis to offer. There are 93 heterodimers plus 112 bp DNA in a 43 Å pitch of the helical capsid. That's a total of about 360 kDa mass (estimating 15 kDa as the subunit mass). The mean area (at 160 Å diameter) of the lipid sheath is 2.1 x 10**4 Å2 per 43Å length. The area occupied by a U-shaped tetraether lipid can be estimated to be about 160 Å2. So there should be 260 lipids in 43 Å length, or just under 100 kDa. That comes to a bit over 20% of the mass, not the 40% the authors estimate from a "crude" calculation or the 43% they cite from the MD. Further, showing a phosphoinositol headgroup lipid in the figure when that modification is not, at least from the text, present in the virus is confusing.

---

## [Author Response]

*Essential revisions:*

*There was overall agreement of the novelty of the finding, but significant concerns emerged both about the computational aspects of the study as well as the style of presentation of the results.*

*1) The lack of sufficient detail about the MD simulations was seen as a major weakness of the manuscript, and led to the concern that something may be amiss in the modeling. Reviewers would need to see snapshots of the 'bilayer' and time-averaged transbilayer distributions of the key molecular features of the lipids discussed in the paper as well as a movie that shows the motions of the lipids and associated waters to be convinced.*

As requested, we have added more details on the MD simulations as well as clear statements of what we can infer directly from the experimental data, what we believe are robust findings in the MD simulations, and relevant uncertainties. We have also added renderings of snapshots for the membrane envelope, a plot of the radial density over time (a key comparison versus experimental data discussed in the paper), movies of the dynamics of twenty randomly selected lipids (sequential lipids in the PDB file) over time and of interfacial water over time as requested. The time-resolved plots show dynamic motion of lipids and water but no drift in the essential features. These plots clearly show that a predominantly horseshoe-conformation envelope is stable and matches the width and placement of the density from cryo-EM with lipid tail composition as per mass spectrometry with a water layer between the envelope and the capsid. We have also clarified in the manuscript (as discussed below) that there is no “bilayer” and that the envelope is the thickness of a single (rather than two) horseshoe-conformation lipids".

*2) Reviewers were concerned that there was not a direct measurement of the lipid content, for example by measuring the buoyant density on a suitable gradient known to provide accurate results for other kinds of viruses.*

In fact, the buoyant density for the AFV1 virions was reported in Bettstetter et al.,2003, using a CsCl gradient. We show in the table below the buoyant density for a number of archaeal, bacterial and eukaryotic dsDNA viruses, both membrane-enveloped and without membranes.

We have now added a sentence to the paper citing the published density and how it is consistent with other membrane-enveloped dsDNA viruses.

**Virus****Family****Lipid**
**membrane****Buoyant density**
**in CsCl, g cm_-3_****Reference**AFV1*Lipothrixviridae*+1.3(Bettstetter et al., 2003)SSV1*Fuselloviridae*+1.3(Quemin et al., 2015)PSV*Globuloviridae*+1.3(Haring et al., 2004)HHPV-1*Pleolipoviridae*+1.27(Roine et al., 2010)*Poxviridae*+1.3(King et al., 2011)PM2*Corticoviridae*+1.28(King et al., 2011)PRD1*Tectiviridae*+1.29(King et al., 2011)SIRV2*Rudiviridae*-1.36(Prangishvili et al., 1999)*Caudovirales*-1.5(King et al., 2011)*Polyomaviridae*-1.35(King et al., 2011)*Papillomaviridae*-1.35(King et al., 2011)*Adenoviridae*-1.31-1.36(King et al., 2011)

*3) It is important to provide a better clarification of the nomenclature and concepts involved in characterizing the membrane, differentiating the "monolayer" vs. "bilayer" aspects of the membrane, including a better representation in figures (e.g. close-up images from Figure 7).*

We have attempted to clarify the text. All references to the AFV1 envelope use “envelope” or “monolayer”, which correctly described the AFV1 envelope. We wonder if the reviewers were referring to the mixed “in/out” orientation of the horseshoe monolayer. We have added explicit clarification of this in the text, describing that the monolayer is distinct from a bilayer because the envelope is only the thickness of one horseshoe-conformation lipid rather than two even if the lipids in this monolayer are not all aligned the same direction.

*4) Title overstates the results, and should reflect that the proposed membrane architecture is derived from a computational model.*

We have altered the title. The new title is “A Novel Membrane Type Envelops a Filamentous Hyperthermophilic Virus”. We have also added a section to the Discussion that explicitly states which of our conclusions derive directly from the experimental data and which from computational models.

*5) The reviewers expressed concern at the long and discursive nature of the manuscript in several places, instead of keeping the text focused on the main results.*

We have tried to edit the text to make the focus clearer.

6) One reviewer had the following analysis to offer. There are 93 heterodimers plus 112 bp DNA in a 43 Å pitch of the helical capsid. That's a total of about 360 kDa mass (estimating 15 kDa as the subunit mass). The mean area (at 160 Å diameter) of the lipid sheath is 2.1 x 10**4 Å2 per 43Å length. The area occupied by a U-shaped tetraether lipid can be estimated to be about 160 Å2. So there should be 260 lipids in 43 Å length, or just under 100 kDa. That comes to a bit over 20% of the mass, not the 40% the authors estimate from a "crude" calculation or the 43% they cite from the MD. Further, showing a phosphoinositol headgroup lipid in the figure when that modification is not, at least from the text, present in the virus is confusing.

We apologize for any confusion regarding the lipid composition. The primary core lipid, as stated in the text, is GDGT-0, which has a molecular weight of 1,301. The common headgroups in the host archaeon are phosphohexose, dihexose, and sulfonated trihexose. Our models used phosphohexose headgroups as the smallest of these, modeled as phosphoinositol (although we also ran PC-GDGT0 models for comparison and obtained similar horseshoe conformations). We have amended the text to specify this.

Using a phosphohexose headgroup, the lipid molecular weight is 1,533. We estimate from the MD simulation ~ 190 lipids in a 43 Å segment of the capsid, fewer than the reviewer estimated. This would yield approximately 290 kDa rather than <100 kDa for the lipid mass. We believe the reviewer must have used a much lower molecular weight estimate for the lipid, accounting for the discrepancy. Or the reviewer simply made a large arithmetic error. The crude calculation referred to in the text is entirely different, and did not derive from estimating the number of lipids. Rather, it involved integrating density and then making the rough assumption that density will be proportional to mass. The remarkable agreement using these two very different approaches is worth noting.